# Effect of Carbon Dots Concentration on Electrical and Optical Properties of Their Composites with a Conducting Polymer

**DOI:** 10.3390/molecules27228000

**Published:** 2022-11-18

**Authors:** Grigorii V. Nenashev, Maria S. Istomina, Roman S. Kryukov, Valeriy M. Kondratev, Igor P. Shcherbakov, Vasily N. Petrov, Vyacheslav A. Moshnikov, Andrey N. Aleshin

**Affiliations:** 1Ioffe Institute, 26 Politekhnicheskaya, 194021 Saint Petersburg, Russia; 2Faculty of Electronics, Saint Petersburg Electrotechnical University LETI, Ul. Professora Popova 5, 197022 Saint Petersburg, Russia; 3Almazov National Medical Research Centre, 2 Akkuratova Street, 197341 Saint Petersburg, Russia; 4Center for Photonics and 2D Materials, Moscow Institute of Physics and Technology, 9 Institutskiy Lane, 141701 Dolgoprudny, Russia

**Keywords:** carbon quantum dots, conductive polymers, composite films, electrical conductivity

## Abstract

CQD/PEDOT:PSS composites were prepared via the hydrothermal method from glucose carbon quantum dots (CQDs) and an aqueous solution of PEDOT:PSS conducting polymer and their electrical and optical properties were investigated. The morphology and structure of these samples were investigated by AFM, SEM, EDX, and EBSD. It was found that the CQDs and CQD/PEDOT:PSS composites had a globular structure with globule sizes of ~50–300 nm depending on the concentration of PEDOT:PSS in these composites. The temperature dependence of the resistivity was obtained for the CQD/PEDOT:PSS (3%, 5%, 50%) composites, which had a weak activation character. The charge transport mechanism was discussed. The dependence of the resistivity on the storage time of the CQD/PEDOT:PSS (3%, 5%, 50%) composites and pure PEDOT:PSS was obtained. It was noted that mixing CQDs with PEDOT:PSS allowed us to obtain better electrical and optical properties than pure CQDs. CQD/PEDOT:PSS (3%, 5%, 50%) composites are more conductive composites than pure CQDs, and the absorbance spectra of CQD/PEDOT:PSS composites are a synergistic effect of interaction between CQDs and PEDOT:PSS. We also note the better stability of the CQD/PEDOT:PSS (50%) composite than the pure PEDOT:PSS film. CQD/PEDOT:PSS (50%) composite is promising for use as stable hole transport layers in devices of flexible organic electronics.

## 1. Introduction

Carbon quantum dots (CQDs) are an advanced and unique class of modern nanomaterials that are attracting considerable attention due to their electrical and optical properties [1,2]. This new class of nanomaterials was first obtained by purification of single-walled carbon nanotubes in 2004 [3] and then by laser ablation of graphite powder and cement in 2006 [4]. Since being obtained, CQDs have aroused rapidly growing and significant interest in global academic and engineering circles, which is noted in the substantial growth of scientific articles published on this topic [5]. Over the past few years, significant progress has been made in the synthesis and application of CQDs [6]. Compared to traditional semiconductor quantum dots and organic dyes, CQDs are promising materials for use in electronic and optoelectronic applications due to their tunable photoluminescence, water-solution stability and high solubility, robust chemical inertness, and ease of surface modification [7]. The excellent biological properties of CQDs, such as low toxicity and good biocompatibility, make it possible to use them in bio-imaging, biosensors, and drug delivery [6,8]. The photoluminescent (PL) properties of CQDs open up a promising roadmap for use in sensor applications [9]. We can especially highlight the physical processes in fluorescent systems, such as energy transfer, fluorescence quenching, and the sensitivity of fluorescence spectra, pointing to the widespread use of CQDs in bio- and chemical probing [10].

PEDOT:PSS is a polymer complex consisting of a polythiophene-derivative poly(3,4-ethylenedioxythiophene) (PEDOT) and a dielectric-component poly(4-styrenesulphonate) (PSS) (PEDOT/PSS). This conductive polymer currently plays an important role in organic electronic devices. It is widely used as a conductive and to inject p-type layers in polymer and composite (organic–inorganic) LEDs, field-effect transistors, solar cells, conductive screens, and coatings. Due to the molecular weight of the components of the complex—the conductive conjugate polymer PEDOT (1000–2500) and the PSS chain (~400,000)—the PSS chain with PEDOT oligomer forms a quasi-one-dimensional structure, the properties of which determine the mechanism of transport of charge carriers in the PEDOT/PSS complex. PEDOT/PSS is very technologically advanced and makes it easy to carry on various substrates in the form of conductive films, but the main disadvantage of PEDOT:PSS is thermal degradation of the electrical conductivity.

The optical properties of CQDs are currently being investigated in sufficient detail [7,11]. There is some research on the influence of CQDs on the optical and electrical properties of polyvinylidene fluoride polymer for optoelectronic applications [12]. Similar studies on the effect of the material on the electrical properties have also been carried out for carbon nanotubes and graphene/PEDOT:PSS composite [13,14]. However, the electrical properties of CQDs films and their composites with semiconducting polymer PEDOT:PSS have not been studied in detail to date. Recently, we reported the first results on the transport properties of composite-film CQDs and PEDOT:PSS [15]. In this article, we describe the synthesis CQDs using the hydrothermal method from glucose and prepare CQDs/PEDOT:PSS composites in different concentrations of CQDs, as well as study the effect of CQD concentration on electrical and optical properties of their composites with semiconducting polymer PEDOT:PSS in order to determine the features of the morphology and optical and electrical properties to clarify the charge-carrier transport mechanism in such structures.

## 2. Results and Discussion

Carbon quantum dot/PEDOT:PSS (CQDs/PEDOT:PSS) composites with a 3%, 5%, and 50% of the content of the PEDOT:PSS was synthesized via the hydrothermal method. Glucose (Vecton, Russia) and an aqueous solution of PEDOT:PSS (Sigma Aldrich) were used as precursors [16]. The chemical process of glucose transition to CQDs during hydrothermal synthesis and PEDOT:PSS molecular structure is shown in Figure 1.

The characterization of CQD/PEDOT:PSS composite sizes was carried out using a particle-size analyzer Zetasizer Nano ZS (model ZEN3600, Malvern Instruments, Worcestershire, UK) (Figure 2a–c). They ranged in size from 20 to 90 nm, from 90 to 900 nm, and from 30 to 100 nm, with peak sizes of 40, 110, and 50 nm, respectively, for CQDs/PEDOT:PSS (3%, 5%, and 50%, respectively). The CQDs/PEDOT:PSS (5%) sample also had a second peak at 500 nm.

The morphology of the CQDs and CQDs/PEDOT:PSS composites was studied by AFM for PEDOT:PSS concentrations of 3%, 5%, and 50%. The results of AFM studies of pure CQDs from our previous work [15] indicate that the CQD films had a developed surface with a heterogeneous morphology, which was characterized by the presence of globules with pronounced boundaries. In this case, the average grain diameter for the CQD films varied in the range of ~50–100 nm, and their height was ~50 nm, which is comparable with the dimensions of the CQDs. The morphology of the CQD/PEDOT:PSS (3%, 5%, 50%) composites turned out to be inhomogeneous and rough (Figure 3a–c) and was more complex in comparison with pure CQDs. Carbon nanoparticles had a globular form with average sizes of 150 nm in the case of CQDs/PEDOT:PSS (3%), 250 nm in the case of CQDs/PEDOT:PSS (5%), and 50–100 nm in the case of CQDs/PEDOT:PSS (50%). AFM imaging revealed a regular pattern on the CQD/PEDOT:PSS (3%, 5%, 50%) surfaces consisting of height peaks and valleys. The nanoparticle heights were 2–4 nm, 1–3 nm, and 0.5–1 nm for the CQDs/PEDOT:PSS (3%, 5%, and 50%, respectively). The number of nanoparticles increased, but their height decreased with increasing content of PEDOT:PSS in the synthesis-precursor solution, which may have indicated the effect of PEDOT:PSS in the formation of nanoparticles.

The CQDs and CQD/PEDOT:PSS (3%, 5%, 50%) composites were studied via scanning electron microscope Zeiss Supra 25 (SEM, Carl Zeiss AG, Oberkochen, Germany) equipped with an energy-dispersive X-ray and an electron backscatter diffraction detector (EDX and EBSD, respectively, National Instruments, Austin, TX, USA). The SEM images are shown in Figure 4. As is clearly seen in the SEM images, which are shown in Figure 4a,b, there was a slight visual difference in the morphology of the study samples of CQDs and CQD/PEDOT:PSS (3%), but the morphology of CQD/PEDOT:PSS (5%) and CQD/PEDOT:PSS (50%) differed from all studied samples (Figure 4c,d). The CQD/PEDOT:PSS composite material had an equal distribution of all components without defects like lumps. Shedding of layers and wrinkles was observed at 5% and 50%, especially at 5%.

Figure 5a presents the EDX spectra of the CQDs and CQD/PEDOT:PSS (3%, 5%, 50%) composites on Si(100) substrate. Here we observed practically equal intensity of peaks, which can be attributed to the signal from the O atoms, for all spectra besides that of 5%. The increase in the O-peak intensity when PEDOT:PSS was embedded into the compound can be connected to the hydrophilic properties of the polymer. There was a low-intensity high-energy S-peak for PEDOT:PSS with a concentration of 50%, which is associated with the PEDOT:PSS and was not found in other concentrations [17,18]. The C-related peak, which also occurred due to carbon collection under the electron beam, was not analyzed due to the high lightness of the chemical element and the poor resolution of the detector near the beryllium window. There were no peaks other than those presented in the high-energy region.

Figure 5b represents the EBSD map of the CQD near-surface area. There were features in the diffraction patterns of backscattered electrons that allowed us to estimate the phase composition of the CQDs as mainly graphite with space groups of Pban (50), Pcca (54) [19], Cmmm (65) [19], and P6_3_mc (186) [20]. The appearance of such graphite clusters can be associated with the agglomeration of colloidal particles in the dry layer.

The FTIR reflectance spectra of the CQD/PEDOT:PSS film are shown in Figure 6. A strong band at 3278 cm^−^^1^ arose from the stretching of the hydroxyl groups. The bands at 2883 cm^−^^1^ and 800 cm^−^^1^ can be assigned to the C–H stretching mode and C–H bending mode. The presence of hydroxyl and carboxyl groups imparts excellent water solubility and stability for various biomedical applications [21]. Bands at 1553 (C=C), 1256 (C–C), and 933 cm^−^^1^ (S–O) originated from the thiophene ring in PEDOT [22]. The vibrations at 1179 cm^−^^1^ were due to the C-O-C bond stretch in the ethylenedioxy group [23]. A band at 1553 cm^−^^1^, typical for free amino groups, was observed, which confirms the N-doping of the carbon core [24]. The bands at 3278, 2883, and 800 cm^−^^1^ were associated with CQDs and the bands at 1256, 1179, and 938 cm^−^^1^ were associated with PEDOT:PSS. The band at 1553 cm^−^^1^ was associated with both. The presence of bands of the spectrum associated with CQDs and PEDOT:PSS indicates the existence of chemical intermolecular interactions between these components.

Typical PL and absorbance spectra of the CQD/PEDOT:PSS (50%) composite are shown in Figure 7 and the inset in Figure 7, respectively. As seen from these figures, the PL maximum of the CQD/PEDOT:PSS (50%) composite corresponded to a wavelength of 490 nm, after which there was a gradual decrease in the PL intensity. The absorption of the CQD/PEDOT:PSS (50%) composite had a more complex character. As can be seen from the inset in Figure 7, the absorption spectrum had a value close to the maximum at a wavelength of 450 nm to 900 nm at the peaks of 450 nm and 900 nm. The absorbance spectra can be explained by the synergistic effect of interaction between the CQD and PEDOT:PSS. The maximum at 450 nm corresponded to the absorption curve of pure CQDs [25]. The range from 600 nm to 900 nm corresponded to the absorption range of pure PEDOT:PSS [22]. Notably, the synergistic effect enhanced the overall absorbance ability of the composite in comparison with pure CQDs and pure PEDOT:PSS.

The current–voltage (I–V) characteristics of the CQD/PEDOT:PSS (3%, 5%, 50%) composites and pure PEDOT:PSS at forward and reverse bias in linear scales at room temperature are shown in Figure 8a. As can be seen in this figure, all measured samples demonstrated good linearity over the entire range of voltages. In this case, the conductivity value of the composite decreased with a decrease in the concentration of PEDOT:PSS. At the same time, the difference in conductivity between the 3% and 5% CQD/PEDOD:PSS composites was more significant than the difference in conductivity between the 5% and 50% CQD/PEDOT:PSS. The difference in conductivity between the 50% CQD/PEDOT:PSS and pure PEDOT:PSS composites was the least significant. Upon irradiation of samples by a simulator of sunlight with a wavelength in the range of 300–700 nm, no noticeable photocurrent was detected, which indicates that all CQD/PEDOT:PSS composites did not exhibit noticeable photoconductivity in the visible spectral range. The temperature dependence of the I–V characteristics of the CQD/PEDOT:PSS (50%) composites at forward and reverse bias in semi-logarithmic scales is shown in Figure 8b. As can be seen from this figure, the CQD/PEDOT:PSS (50%) composite had a low resistivity at room temperature, which increased with decreasing temperature. The resistivity of the CQD/PEDOT:PSS (50%) composites was ~10^1^ Ω·cm and ~10^2^ Ω·cm for 77 K and 293 K, respectively, which, along with the good linearity of the I–Vs, is a promising characteristic for device applications.

The temperature dependence of the resistivity for the CQD/PEDOT:PSS (50%) sample is shown in Figure 9. This ρ(*T*) dependence had a weak activation character and can be described by the following expression:(1)$$\rho \left(T\right)=\rho_{0}exp\left(E_{a}/k_{B}T\right)$$
where *E_a_* is the activation energy, *T* is the temperature, and *k_B_* is the Boltzmann constant. The activation energy of conduction *E_a_* was calculated from the temperature dependence ρ(*T*) by the following formula: (2)$$E_{a}\left(meV\right)=\left(200\Delta \log\rho \right)/\left(\Delta 1000/T\right),$$
where ρ is the resistivity of the film.

The obtained value of Ea was ∼7.5 meV, which for the CQD/PEDOT:PSS (50%) composite, in our opinion, indicates the hopping conductivity of charge carriers between impurity states inside the band gap. CQD agglomerates were unevenly distributed in the PEDOT:PSS matrix, which led to the presence of surface areas of the PEDOT:PSS conductive polymer free of CQDs, along which, in our opinion, the main transport of charge carriers takes place in such systems. This fact can also explain the I–V dependence in Figure 8a.

The dependence of the resistivity on the storage time of the PEDOT:PSS and CQDs/PEDOT:PSS composites is shown in Figure 10. The CQD/PEDOT:PSS (50%) sample had better stability compared to the other samples and is promising for use in device applications. The CQD/PEDOT:PSS (3%) and CQD/PEDOT:PSS (5%) samples showed a clear degradation, which is expressed in the high resistance values of the samples.

The results of comparing the values of resistivity and conductivity of composite films obtained in our work with similar data for PEDOT:PSS composites with other fillers obtained in the works of other authors are presented in Table 1.

Therefore, the results obtained show that the CQD/PEDOT:PSS composite exhibited lower conductivity than graphene/PEDOT:PSS, carbon nanotube/PEDOT:PSS, and metal-nanoparticle/PEDOT:PSS composites and pure PEDOT:PSS, but allows a stable composite to be obtained that demonstrates stable resistivity for more than 300 days.

## 3. Materials and Methods

### 3.1. Synthesis of CQDs and CQD/PEDOT:PSS Composites

CQD/PEDOT:PSS composites were prepared by the hydrothermal method from glucose (Figure 1a) and an aqueous solution of PEDOT:PSS (Sigma Aldrich) (Figure 1b) with 3%, 5%, and 50% content of the PEDOT:PSS. The CQD solutions were obtained using 2.7 g of glucose, which was dissolved in 15 mL of distilled water. Then, to obtain a 3% CQD/PEDOT:PSS solution, 0.45 mL of CQD solution were taken out and 0.45 mL of PEDOT:PSS solution were added. In the case of the 5% CQD/PEDOT:PSS solution, 0.75 mL of CQD solution were taken out and 0.75 mL of PEDOT:PSS solution were added. For the 50% CQD/PEDOT:PSS solution, 5 mL of CQD solution were taken out and 5 mL PEDOT:PSS solution were added. Next, all these solutions were thoroughly mixed. The resulting solutions were formed by heating in a steel miniautoclave at T = 160 °C for t = 6 h, and after that there were centrifuged at 4000 rpm for 10 min to remove large particles and additionally purified from the chemical reaction byproducts using water dialysis through a dialysis membrane (12 kDa) for 5 days with a daily water replacement. It should be mentioned that all the initial components used during hydrothermal synthesis were aqueous dispersion, which is the reason for their compatibility in the preparation of composite solutions.

### 3.2. Characterization Techniques

The characterization of CQD/PEDOT:PSS composite sizes was carried out using a particle-size analyzer Zetasizer Nano ZS (model ZEN3600, Malvern Instruments, Worcestershire, UK). They ranged in size from 20 to 90 nm, from 90 to 900 nm, and from 30 to 100 nm, with peak size of 40, 110, and 50 nm, respectively for CQDs/PEDOT:PSS (3%, 5%, and 50%, respectively). The CQD/PEDOT:PSS (5%) sample also had a second peak at 500 nm. The morphology of the CQDs and CQD/PEDOT:PSS composites with 3%, 5%, and 50% concentrations was studied using atomic force microscopy (AFM) via SOLVER P47-PRO NT-MDT. The CQDs and CQD/PEDOT:PSS (3%, 5%, 50%) composites were studded via scanning electron microscope Zeiss Supra 25 (SEM, Carl Zeiss AG, Oberkochen, Germany) equipped with an energy-dispersive X-ray and an electron backscatter diffraction detector (EDX and EBSD, respectively, National Instruments, Austin, Texas, USA). Fourier transform infrared (FTIR) spectra of the CQD/PEDOT:PSS film were recorded with an FTIR spectrometer (IR Prestige 21, Shimadzu Scientific Instruments, Kyoto, Japan) in the range of 400–4000 cm^−1^. The absorption and photoluminescence spectra of the CQDs deposited on quartz substrates were recorded using high-sensitivity fiber-optic spectrometer AVANTES—AvaSpec-ULSi2048L-USB2 OEM—operating in the spectral range of 322–1100 nm.

### 3.3. Sample Preparation

To study the electronic properties, CQD/PEDOT:PSS films with different contents of PEDOT:PSS (3%, 5%, 50%) were prepared. Glass plates with an ITO layer (Sigma Aldrich) were used as substrates. The distance between the planar ITO electrodes was ~200 μm, and the width of the electrodes was ~5 mm. Solutions of aqueous dispersion of CQDs and PEDOT:PSS were mixed in the ratio of CQDs/PEDOT:PSS components as follows: CQDs—100%, 0.8: 0.2, 0.5: 0.5, and PEDOT:PSS—100% at 300 K using an ultrasonic mixer Bandelin SONOPULS ultrasonic homogenizer at a frequency of f = 20 kHz for 2 min until a uniform consistency was reached. The resulting solutions were applied onto glass substrates with ITO electrodes by drop casting (the film thickness was measured by AFM and was 0.5 µm). The samples applied to the substrates were dried using a heater with controlled temperature (Heidolph HG 3001 K, Schwabach, Germany) at 100 °C in N_2_ atmosphere for 15 min in an inert box DX-2.

### 3.4. Current–Voltage Characteristics

The current–voltage (I–V) characteristics of the samples placed in the holder of an optical flow through nitrogen cryostat with temperature-stabilization OPTCRYO198 were measured at a direct current in a nitrogen atmosphere in the dark and under illumination with a simulator of sunlight in the temperature range of 77–280 K with a step of 10–20 K using an automated-measurement setup based on a Keithley 6487 picoammeter. The voltage was changed with variable steps in the range of −5 V to +5 V. Electrical contacts to the samples were made of silver wire, which was attached to metal electrodes using a carbon or silver paste (SPI).

## 4. Conclusions

CQDs were synthesized using the hydrothermal method from glucose, composites with a PEDOT:PSS-conducting polymer in different concentrations were obtained, and the electrical and optical properties of all composites were investigated. The morphology of the CQD/PEDOT:PSS (3%, 5%, 50%) composites turned out to be heterogeneous, rougher, and more complex in comparison with the morphology of pure CQDs, which had a developable surface. Surface roughness increased with increasing concentration of PEDOT:PSS. PEDOT:PSS also contributed to the appearance of CQD-free areas, which significantly affected the electrical characteristics of these samples. The I–V characteristics were obtained for pure PEDOT:PSS and CQD/PEDOT:PSS composites, including the temperature I–V characteristic and resistivity-on-temperature characteristic for the CQD/PEDOT:PSS (50%) sample. Mixing CQDs with PEDOT:PSS (1:1) allowed more conductive composite to be obtained than pure CQDs and more stable composite to be obtained than pure PEDOT:PSS and showed better stability than pure PEDOT:PSS films, which is promising for use as stable hole transport layers in devices of flexible organic electronics.

## Figures and Tables

**Figure 1 molecules-27-08000-f001:**
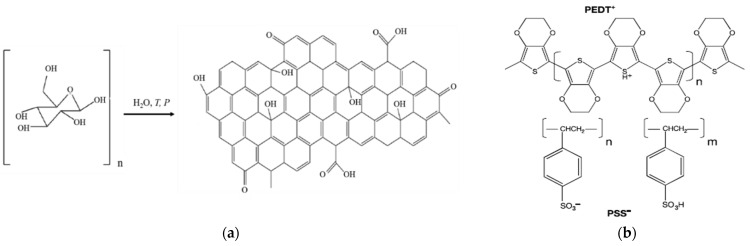
Chemical process of glucose transition to CQDs during hydrothermal synthesis (**a**) and PEDOT:PSS molecular structure (**b**).

**Figure 2 molecules-27-08000-f002:**
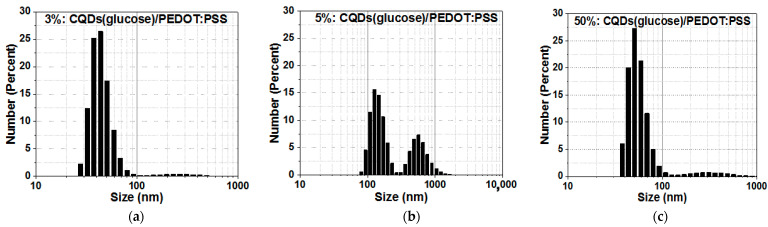
Size distribution of CQDs by number for (**a**) 3%, (**b**) 5%, and (**c**) 50% concentrations of PEDOT:PSS.

**Figure 3 molecules-27-08000-f003:**
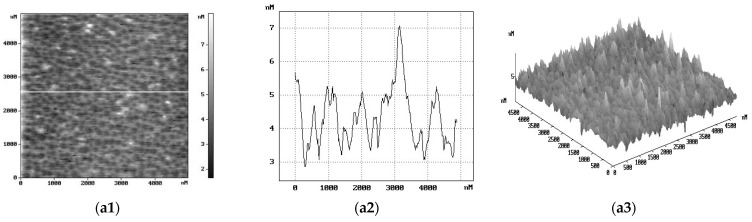
Results of AFM studies of the morphology of (**a**) 3% CQDs/PEDOT:PSS (**a1**–**a3**), (**b**) 5% CQDs/PEDOT:PSS (**b1**–**b3**), and (**c**) 50% CQDs/PEDOT:PSS (**c1**–**c3**) composites.

**Figure 4 molecules-27-08000-f004:**
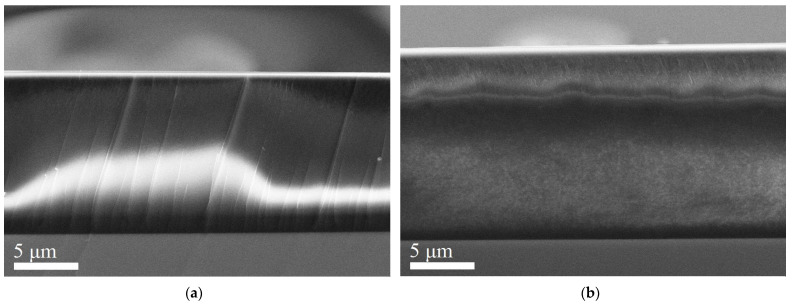
SEM image of (**a**) pure CQDs, (**b**) 3% CQD/PEDOT:PSS composite, (**c**) 5% CQD/PEDOT:PSS composite, and (**d**) 50% CQD/PEDOT:PSS composite coated with the colloidal solution on silicon substrate and annealed at 150 for 30 min.

**Figure 5 molecules-27-08000-f005:**
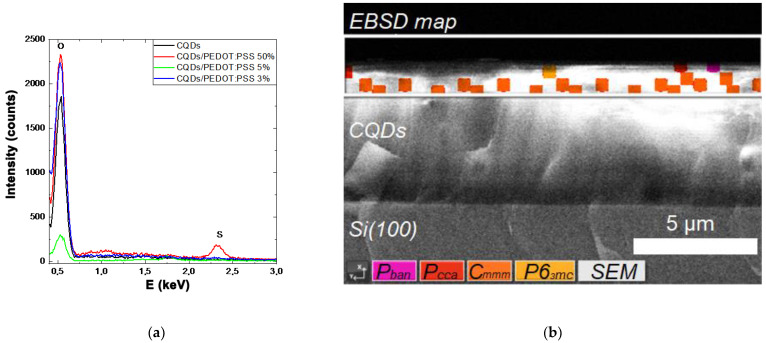
(**a**) EDX spectra of the CQDs and CQD/PEDOT:PSS (3%, 5%, 50%) composites and (**b**) EBSD map of the CQDs.

**Figure 6 molecules-27-08000-f006:**
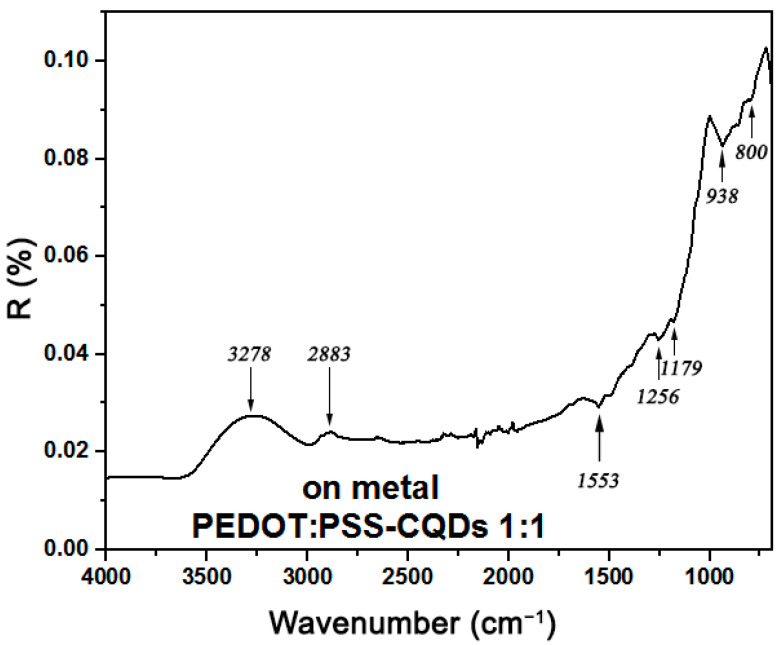
Reflectance FTIR spectra of the CQD/PEDOT:PSS film on metal.

**Figure 7 molecules-27-08000-f007:**
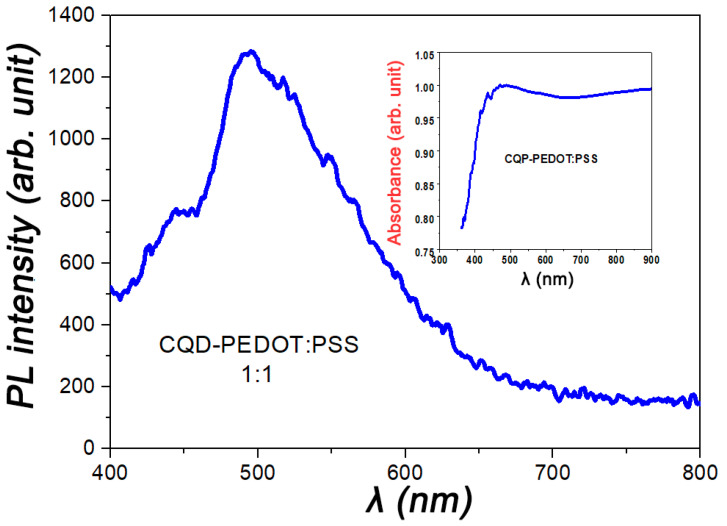
PL and absorbance (inset) and spectra for 50% CQD/PEDOT:PSS composite film.

**Figure 8 molecules-27-08000-f008:**
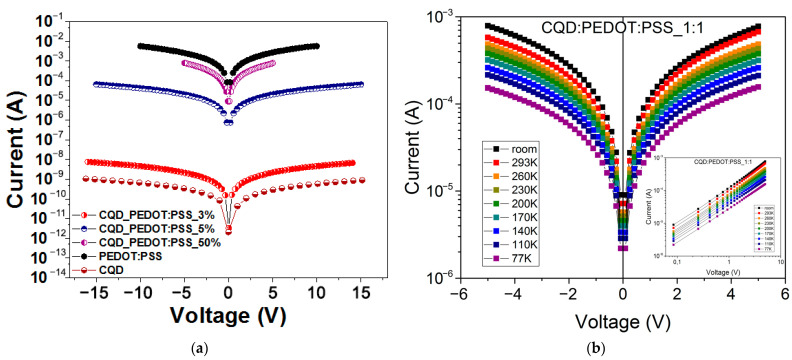
Current–voltage characteristics of the CQD/PEDOT:PSS composites and pure PEDOT:PSS in the dark (**a**). CQD/PEDOT:PSS (50%) composites at forward and reverse bias in semi-logarithmic scales at different temperatures (77–300 K) (**b**).

**Figure 9 molecules-27-08000-f009:**
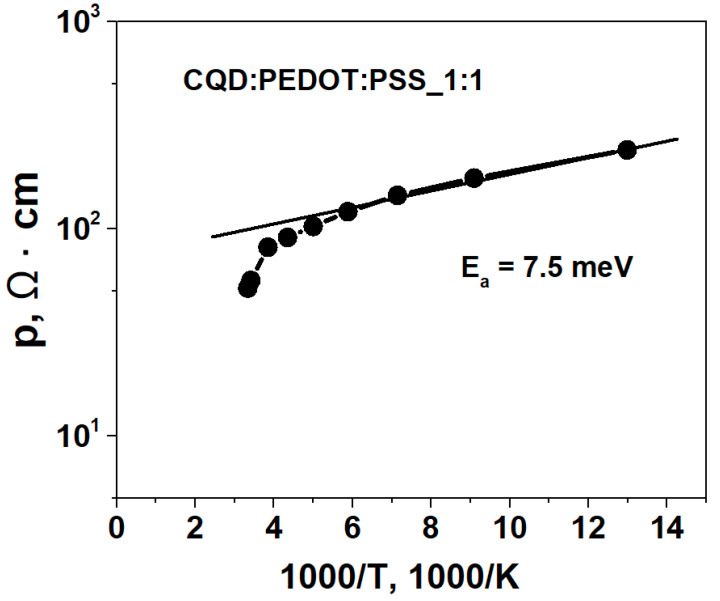
Dependence of the resistivity on the inverse temperature of the CQD/PEDOT:PSS (50%) composite.

**Figure 10 molecules-27-08000-f010:**
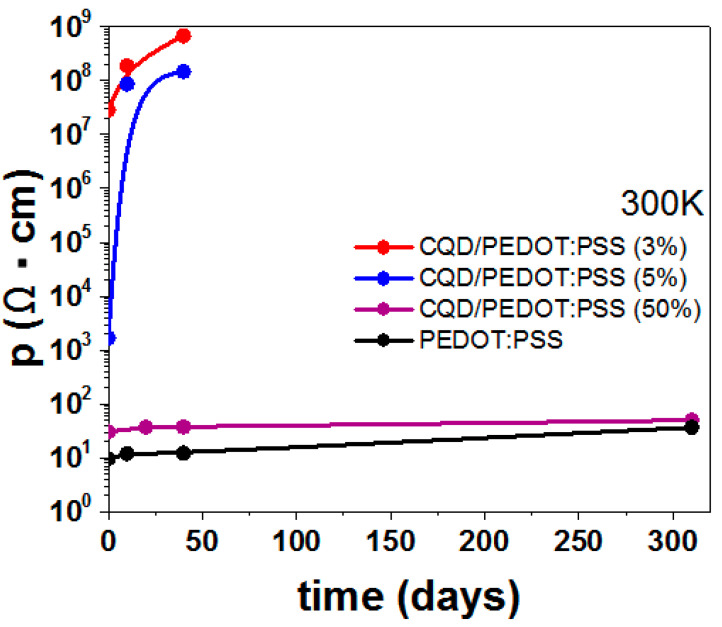
Dependence of the resistivity on the storage time of the PEDOT:PSS and CQD/PEDOT:PSS composites with different contents of CQDs and PEDOT:PSS.

**Table 1 molecules-27-08000-t001:** Comparison of resistivity and conductivity of carbon-based materials.

	Resistivity, Ω cm	Conductivity, S/cm	Reference
Graphene/PEDOT:PSS (0.02 wt%)	6.25	0.16	[13]
Graphene/PEDOT:PSS (0.1 wt%)	0.77	1.29	[13]
Graphene/PEDOT:PSS (0.5 wt%)	0.39	2.58	[13]
Graphene/PEDOT:PSS (1 wt%)	0.16	6.24	[13]
Carbon nanotubes/PEDOT:PSS/Au/polymer (60%, 20%, 10%, 10%)	0.0004	2400	[14]
Carbon nanotubes/PEDOT:PSS/Au/polymer (60%, 15%, 15%, 10%)	0.0002	6000	[14]
Metal nanoparticles (Au)/PEDOT:PSS	0.15	6.67	[26]
Carbon quantum dots/PEDOT:PSS	30.03	0.03	This article
Pure PEDOT:PSS	9.54	0.1	This article

## Data Availability

Data presented in this study are available on request from the corresponding author.

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
