# Peer review of "Effect of Carbon Dots Concentration on Electrical and Optical Properties of Their Composites with a Conducting Polymer"

_molecules, 2022, doi:10.3390/molecules27228000_

Round 1

Reviewer 1 Report

In this work, the authors have prepared of CQDs/PEDOT:PSS for  use as stable hole transport layers in devices of flexible organic electronics. The authors shows characterization results to indicating specific morphology and functional group of the composite along with electrical and optical properties. However, I think there is still a lot of critical analysis lacking, so it is advisable for the authors to add more results. The authors should make major revision to this research based on the recommendations below for consideration before publication.

1.    As the authors shown the AFM results I think that only surface morphology of  CQDs/PEDOT:PSS composites coated on substrate. Scanning Electron Microscope (SEM) and Transmission Electron Microscope (TEM) are suitable morphology characterization technique for nanocomposite to clarify bulk composite that synthesized. So, I recommend adding SEM or TEM study for 3% 5% and 50% of CQDs/PEDOT:PSS composites in morphology part.

2.    In FTIR result, it is indicating functional group of PEDOT:PSS but it is not clarify for CQDs. The authors should consider adding more characterize all composite by Raman spectroscopy UV-vis and XRD to confirm that carbon dots is completely composite.

3.    How is the composites film thickness that can be prepared and what method is used to prepare the film? (In the methods it was indicated that it was drop casting or spin coating). Because the film thickness has a great effect on the electrical resistance, the authors should clearly specify the thickness of the prepared film. The authors should consider use 4-point probe method to measure resistance of the composites film.

4.    The authors should consider adding comparisons of this research to similar previous studies, especially with carbon-based materials such as graphene/PEDOT:PSS, carbon nanotubes/PEDOT:PSS and other like metal nanoparticles/PEDOT:PSS to confirm the improved development of this research.

Author Response

Please find a point-by-point response to the reviewer 1 in the attached file.

Reviewer 2 Report

Please explain what does PEDOT:PSS stand for. 

Please clearly state, from the very beginning, and then use it in an uniform throughout the whole manuscript, what does the 3%, 5% and 50% from the CQDs/PEDOT:PSS composite refer to: CQDs or PEDOT:PSS? Just an example of contradiction: In Chapter 3.1, it is stated that "CQDs/PEDOT:PSS composites were prepared by hydrothermal method from glucose (Figure 1a) and aqueous solution of PEDOT:PSS (Sigma Aldrich) (Figure 1b) with a 3%, 5% and 50% content of the PEDOT:PSS", leading to the hypothesis that 3%, 5% and 50% refer to the content of PEDOT:PSS in the composite (as a consequence, the whole AFM and FTIR analysis in Chapter 2 is logically performed by comparison with pure CDQs). However, in Chapter 3.3, it is stated that "To study the electronic properties, the CQDs/PEDOT:PSS films with different content of CQDs were prepared", indicating that, actually, 3%, 5% and 50% refer to the content of CQDs in the composite (as a consequence, the whole IV characteristics analysis in Chapter 2 is logically performed by comparison with pure PEDOT:PSS). All in all, this is the big issue the authors should address in the revised version of the manuscript: what do they want to improve in terms of optical and electrical properties: pure PEDOT:PSS or pure CQDs?

By looking at Figure 6a, it seems that pure PEDOT:PSS have better conduction properties when compared to the composites studies. Then why should one choose working with any of the composites instead of employing PEDOT:PSS? Please also add on the that figure the I/V characteristic for pure CQDs.

Author Response

Please find a point-by-point response to the reviewer 2 in the attached file.

Round 2

Reviewer 1 Report

The authors should make minor revision to this research based on the recommendations below for consideration before publication.

11. The authors should consider adding comparisons table of this research to similar previous study articles (e.g. conductivity, film resistance), especially with carbon-based materials such as graphene/PEDOT:PSS, carbon nanotubes/PEDOT:PSS and other like metal nanoparticles/PEDOT:PSS to confirm the improved development of this research.

Author Response

Please find a point-by-point response to the reviewer's comments in the attached file

Reviewer 2 Report

Please revisit the English in the whole manuscript.

Author Response

(The authors gave the same response as above.)
